# Analysis of Tea Plantation Suitability Using Geostatistical and Machine Learning Techniques: A Case of Darjeeling Himalaya, India

**Netrananda Sahu** [1,*] , **Pritiranjan Das** [1,2] , **Atul Saini** [3] , **Ayush Varun** [1] , **Suraj Kumar Mallick** [4] , **Rajiv Nayan** [5] , **S. P. Aggarwal** [5] , **Balaram Pani** [6] , **Ravi Kesharwani** [1] **and Anil Kumar** [1]

[1] Department of Geography, Delhi School of Economics, University of Delhi, New Delhi 110007, India; pdas@geography.du.ac.in (P.D.); ayushdsedu@gmail.com (A.V.); ravikesharwani303@gmail.com (R.K.); anilnaindu@gmail.com (A.K.)

[2] Department of Geography, Shaheed Bhagat Singh Evening College, University of Delhi, New Delhi 110017, India

[3] Delhi School of Climate Change & Sustainability, Institution of Eminence, University of Delhi, New Delhi 110007, India; dsccs.asaini@ioe.du.ac.in

[4] Department of Geography, Shaheed Bhagat Singh College, University of Delhi, New Delhi 110017, India; surajkumarmallick@sbs.du.ac.in

[5] Department of Commerce, Ramanujan College, University of Delhi, New Delhi 110019, India; rajivnayan@ramanujan.du.ac.in (R.N.); spa15_dbce@yahoo.com (S.P.A.)

[6] Department of Chemistry (Environmental Science), Bhaskarcharya College of Applied Science, University of Delhi, New Delhi 110075, India; balaram.pani@bcas.du.ac.in

* Correspondence: nsahu@geography.du.ac.in

**Abstract:** This study aimed to identify suitable sites for tea cultivation using both random forest and logistic regression models. The study utilized 2770 sample points to map the tea plantation suitability zones (TPSZs), considering 12 important conditioning factors, such as temperature, rainfall, elevation, slope, soil depth, soil drainability, soil electrical conductivity, base saturation, soil texture, soil pH, the normalized difference vegetation index (NDVI), and land use land cover (LULC). The data were normalized using ArcGIS 10.2 and the models were calibrated using 70% of the total data, while the remaining 30% of the data were used for validation. The final TPSZ map was classified into four different categories: highly suitable zones, moderately suitable zones, marginally suitable zones, and not-suitable zones. The study revealed that the random forest (RF) model was more precise than the logistic regression model, with areas under the curve (AUCs) of 85.2% and 83.3%, respectively. The results indicated that well-drained soil with a pH range between 5.6 and 6.0 is ideal for tea farming, highlighting the importance of climate and soil properties in tea cultivation. Furthermore, the study emphasized the need to balance economic and environmental considerations when considering tea plantation expansion. The findings of this study provide important insights into tea cultivation site selection and can aid tea farmers, policymakers, and other stakeholders in making informed decisions regarding tea plantation expansion.

**Keywords:** tea plantation; site suitability; random forest; logistic regression; machine learning; Darjeeling

## 1. Introduction

Darjeeling tea (*Camellia sinensis* L.) is world-famous for its unique taste [1]. It was the first Indian product to obtain a geographical indication (GI) tag. The tea plantations of Darjeeling are the major contributors to the region's economy. The expansion of the tea plantation areas is, therefore, essential for long-term socioeconomic sustainability [2]. The increased demand for Darjeeling tea and the limited land available for tea plantations have presented difficulties in recent times. Along with socioeconomic development, tea garden development plays a critical role in attaining self-sufficiency for the dependent population [3].

Land suitability analysis (LSA) is a technique for determining the inherent and potential capabilities, as well as the suitability, of various goals [4]. Geospatial data are successfully investigated using geographic information system (GIS)-based multi-criteria evaluation (MCE) methodologies to permit rigorous and flexible land suitability analysis, which include the analytical hierarchical process (AHP), the frequency ratio (FR), the weight of evidence (WoE), and the evidence belief function (EBF) [5]. Physical criteria such as geology, soil characteristics, geomorphology, atmospheric conditions, vegetation conditions, and economic and sociocultural circumstances are considered when evaluating a site's suitability [6]. The integration of geospatial techniques with the multi-criteria decision-making (MCDM) method has been employed in numerous studies to effectively address intricate challenges pertaining to land management.

These techniques are extensively used in LSA to identify the potential lands for watershed management [7], plantation [3], and agriculture [8–12]. Jayansinghe et al. (2020) assessed the land suitability for tea crops in Sri Lanka using the AHP and the decision-making trial and evaluation laboratory (DEMATEL) model, wherein all the conditioning factors were integrated to generate a land suitability map [13]. In their study on sustainable tea-production-suitability areas in Bangladesh, Das et al. (2020) used phenological datasets from remote sensing and geospatial datasets of soil-plant biophysical properties, and based their findings on expert opinions [14]. The classification of the suitability zones was finally determined using a weighted overlay spatial analysis technique. This method included integrating reclassified raster layers for all important criteria, as well as the results of the multi-criteria analysis. In their study on tea suitability along the Laos–China border, Chanhda et al. (2010) used a combination of multi-criteria analysis and system dynamics techniques to analyze forest land utilization and land suitability, with the goal of anticipating future land use patterns for tea cultivation. [15]. Kamkar et al. (2014) opined that selecting locations for agricultural land use necessitates taking into account geophysical constraints, different topological conditions, and climatic conditions [16]. Accordingly, it is important to take into account current and accurate land use/land cover (LULC), as well as other geographical, environmental, and environmental thematic layers, when determining the best tea-plantation suitability.

New-age machine learning techniques have also been employed for land suitability assessment (LSA) studies. In their study on wheat production in Iran, Fereydoon et al. (2014) employed a two-class support vector machine (SVM) model, wherein 22 models of soil profile information were used, including relief, slope, precipitation, temperature, calcium carbonate content, organic carbon content, pH, and gypsum content [17]. The models were tested on a nonlinear class boundary that provided results of R = 0.84 and RMSE = 3.72. Dahikar and Rode (2014) demonstrated the employability of an artificial neural network (ANN) for ascertaining the cropland suitability zones [18]. They used attributes such as pH, potassium content, sulfur content, manganese content, iron content, soil depth, temperature, rainfall, and humidity. Mokarram (2015) employed several machine learning algorithms such as Bagging, adoptive boosting (AdaBoost), and RotForest for wheat land suitability analysis [19]. The thematic layers included in the study included topography, alkalinity, salinity, soil texture, calcium carbonate content, soil depth, pH, gypsum content, and wetness. The study found that the RotBoost algorithm provided the highest-accuracy score.

In their study on the suitability evaluation of tea cultivation in the Xinming Township in Huangshan City, Anhui Province, China, Xing et al. (2022) compared the following machine-learning-based models: logistic regression (LR), extreme-gradient boosting (XGBoost), AdaBoost, gradient-boosting decision tree (GBDT), random forest (RF), Gaussian Naïve Bayes (GNB), and multilayer perceptron (MLP). These models were employed as computational models to ascertain the precision of weight calculations for the evaluation factors [20]. Xing et al. selected twelve factors and, using the RF model, calculated the evaluation factors' weights and obtained suitability evaluation results.

Wei and Zhou (2023) compared different machine-learning and deep-learning models for evaluating suitable areas for the best tea in Yunnan, China. For the computation

of the evaluation variables, climate, terrain, and green-cropping system variables were used [21]. Six machine-learning models were used in the study. Among all the models, the FA + ResNet model demonstrated the best performance, with an accuracy score of 0.94 and a macro F1 score of 0.93.

In the current study, due to the study area's great elevation variations and the abundance of natural resources, there is very little land that is accessible for tea-garden expansion. In land-suitability mapping, the priority is to identify suitable lands with the highest productivity and the lowest input. Therefore, the analysis of agricultural land's suitability is a reliable and appropriate technique. The primary goal of the current study was to use a random forest and logistic regression model to examine the tea-plantation-suitability zones (TPSZs) in the Darjeeling district.

## 2. Study Area

The strategic geographical position of Darjeeling (Figure 1) holds significant importance, as it shares borders with Sikkim to the north, Bhutan to the east, and Nepal to the west. This configuration establishes Darjeeling as a crucial region, encompassing international boundaries and interstate border areas. Darjeeling tea is known for its unique flavor; it is often referred to as the "champagne of teas".

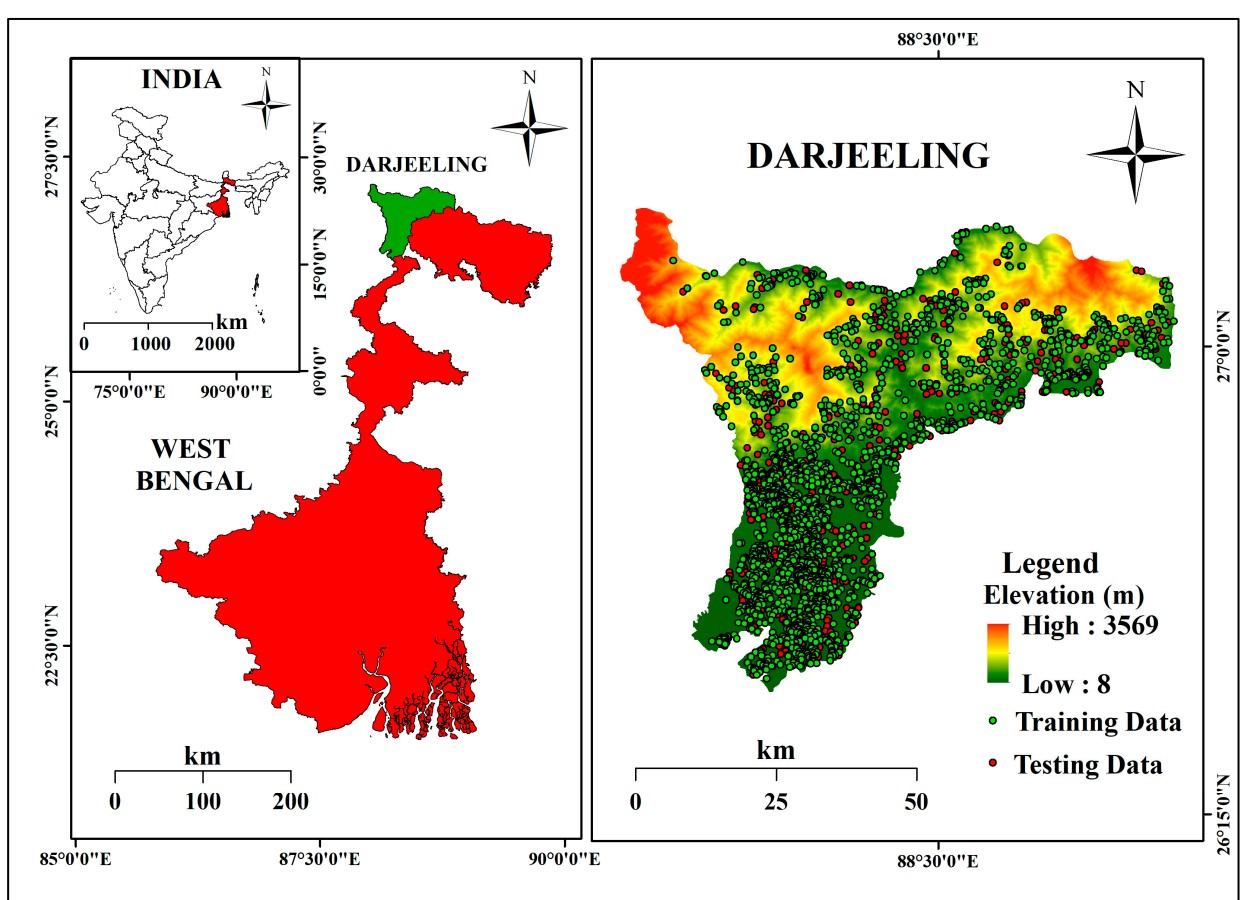

**Figure 1.** Location map of the study area.

Darjeeling's climate is suitable for tea growth. The environment in the region is cool and moist, ideal for growing tea. The geographical extension of this area is 26°27′ N to 27°13′ N and 87°59′ E to 88°53′ E. The average annual temperature exhibits variation, ranging from 24 °C in the low-lying areas to less than 12 °C at higher elevations. In the summer season, temperatures on the ridge can reach approximately 16 °C to 17 °C, while during winter, they tend to drop to around 5 °C to 6 °C. The southern slope of the ridge

receives much higher rainfall (4000–5000 mm) than the leeward sides (2000–2500 mm). Darjeeling tea is cultivated within the elevation range of 3000 to 7000 feet above sea level. This altitude contributes to the tea's distinct fragrance and aroma. Darjeeling's soil is rich in minerals, which contributes to the tea's flavor. Darjeeling tea is traditionally farmed on tiny, family-owned tea estates. To assure the best quality, the tea is grown using traditional methods and the leaves are hand-picked.

Darjeeling tea is harvested four times a year, with each season yielding a unique flavor character. The first flush, which occurs in March and April, produces a light and delicate tea. The second flush, which occurs in May and June, produces a fuller-bodied tea with a musky aroma. The monsoon flush, which occurs in July and August, produces a tea with a robust flavor. The autumn flush, which occurs in October and November, produces a tea with a milder flavor. Overall, Darjeeling tea cultivation is a highly specialized process that requires skill and expertise [22]. The result is a tea that is highly prized and sought after by tea connoisseurs around the world.

## 3. Materials and Methods

### *3.1. Database*

To carry out the TPSZ analysis, different types of data were used. The data were acquired from different sources (Table 1). The LULC map for 2022 was prepared using Landsat 8 OLI data in Google Earth Engine [23], using the support vector machine (SVM) method [24,25]. The LULC map was used for TPSZ analysis after validation by the Kappa co-efficient method [26,27]. Tea-plantation sample points were collected randomly using the LULC map and field surveys.

**Table 1.** Different data sources for the tea-plantation-suitability analysis (accessed on 5 January 2023).

| Thematic Layers | Dataset | Resolution | Sources |
|---|---|---|---|
| Rainfall | CRU | 0.5 km × 0.5 km | https://crudata.uea.ac.uk |
| Temperature | CRU | 0.5 km × 0.5 km | https://crudata.uea.ac.uk |
| Elevation | SRTM DEM | 30 m × 30 m | https://earthexplorer.usgs.gov |
| Slope | Calculated from SRTM DEM | 30 m × 30 m | https://earthexplorer.usgs.gov |
| Soil depth | BHUVAN | 5 km × 5 km | https://bhuvan-app3.nrsc.gov.in |
| Soil drainability | ISRIC | 0.5° × 0.5° | https://data.isric.org |
| Soil electrical conductivity | ISRIC | 0.5° × 0.5° | https://data.isric.org |
| Base saturation | ISRIC | 0.5° × 0.5° | https://data.isric.org |
| Soil texture | ISRIC | 0.5° × 0.5° | https://data.isric.org |
| Soil pH | Soil grids | 250 m × 250 m | https://soilgrids.org |
| NDVI | Landsat 8 OLI | 30 m × 30 m | https://earthexplorer.usgs.gov |
| LULC | Landsat 8 OLI | 30 m × 30 m | https://earthexplorer.usgs.gov |

### *3.2. Conditioning Factors for Tea Cultivation*

#### 3.2.1. Rainfall

Rainfall is one of the most important factors that affect tea cultivation, as it plays a vital role in determining the growth and yield of tea plants. Tea plants require a specific amount of rainfall for optimal growth and yield, which varies depending on the stage of growth. During the vegetative stage, tea plants require regular and adequate rainfall, typically ranging from 1500–2500 mm annually, for the growth of new leaves and shoots. However, during the reproductive stage, tea plants require less rainfall, ranging from 1000–1500 mm annually (Figure 2A), to support the flowering and development of quality shoots [28,29]. Insufficient rainfall can lead to water stress, reduced photosynthesis, and poor growth and yield of tea plants. On the other hand, excessive rainfall can lead to waterlogging, soil erosion, and the spread of diseases, resulting in poor-quality tea leaves.

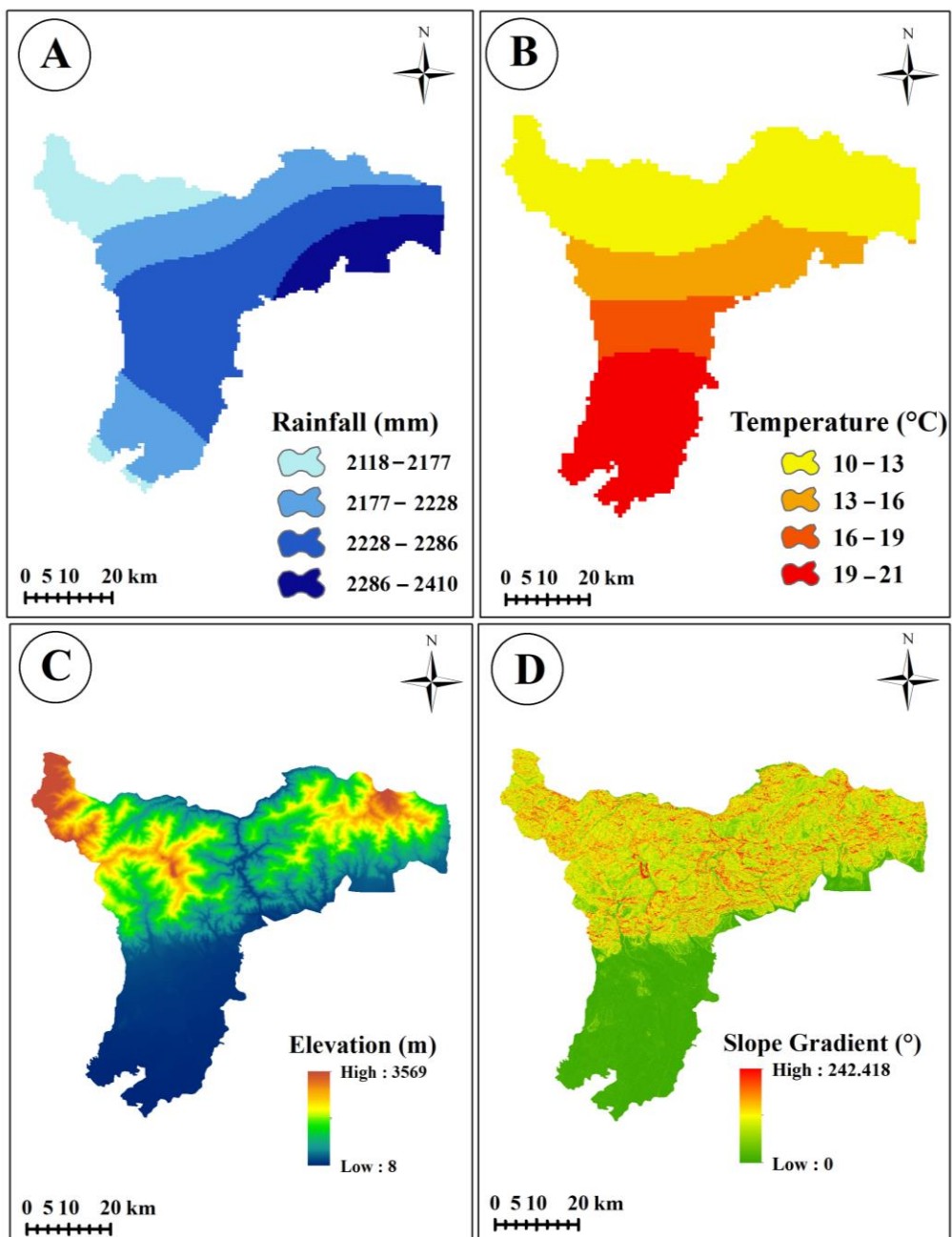

**Figure 2.** Suitability thematic layers for tea plantations: (**A**) rainfall, (**B**) temperature, (**C**) elevation and (**D**) slope.

### 3.2.2. Temperature

Temperature is one of the most important factors that affect tea cultivation, as it affects the growth, development, and quality of tea plants. According to a study [30], tea plants have a specific temperature range for optimal growth and yield, which varies depending on the stage of growth. During the vegetative stage, temperatures ranging from 20–30 °C are ideal for the growth of tea plants. However, during the reproductive stage, temperatures ranging from 10–20 °C are considered ideal for flowering and the development of quality shoots (Figure 2B).

Extreme temperatures, either too high or too low, can have adverse effects on tea plants, resulting in stunted growth, reduced yield, and poor quality of tea leaves. For instance, temperatures above 35 °C can result in excessive transpiration, leading to water

stress and reduced photosynthesis, while temperatures below 10 °C can result in frost damage [31].

### 3.2.3. Elevation

Elevation is another critical factor that affects tea cultivation, as it affects the growth, development, and quality of tea plants. Tea plants have a specific elevation range for optimal growth and yield, which varies depending on the climatic conditions of the region. Tea plants grow best at elevations ranging from 600 to 2000 m above sea level (Figure 2C), with the ideal elevation varying depending on the region [32]. Higher elevations provide cooler temperatures and lower humidity, which result in slower growth and better-quality tea leaves with more complex flavors and aromas.

On the other hand, lower elevations provide warmer temperatures and higher humidity, which result in faster growth and higher yields but may also result in lower-quality tea leaves with a less-complex flavor profile.

### 3.2.4. Slope

Slope is another important factor that affects tea cultivation, as it affects the water and nutrient availability of tea plants. Slope plays a significant role in tea cultivation, as it affects soil erosion, water-holding capacity, and nutrient availability. Gentle slopes (less than 20 degrees) are considered ideal for tea cultivation, as they promote water infiltration, reduce soil erosion, and provide sufficient water and nutrients to tea plants. In contrast, steep slopes (greater than 20 degrees) are not suitable for tea cultivation (Figure 2D), as they promote soil erosion, reduce water infiltration, and result in poor growth and yield of tea plants [33].

In addition, the slope affects the microclimate of a tea plantation, with south-facing slopes receiving more sunlight and heat, resulting in faster growth and higher yields. However, the south-facing slopes may also experience higher water stress, requiring irrigation to maintain optimal soil-moisture levels.

### 3.2.5. Soil Depth

Tea plants require well-drained soil with good water-holding capacity, and the depth of the soil determines the root system's development and the availability of nutrients and water. The depth of the soil significantly influences the growth and yield of tea plants [34]. Tea plants grown in deeper soils have higher shoot growth, greater leaf area, and higher yield than plants grown in shallow soils (Figure 3A). Researchers have attributed this to the greater availability of soil moisture and nutrients in deeper soils, which promote better root development and plant growth. Soil depth significantly affects the accumulation of catechins and caffeine in tea leaves. Tea plants grown in deeper soils have higher concentrations of these compounds, which are important contributors to the taste and health benefits of tea [35].

Soil depth is crucial in tea cultivation, as it affects the growth, yield, and quality of tea plants. Tea farmers should consider the depth of the soil when selecting sites for tea cultivation, and implement appropriate soil management practices to ensure optimal plant growth and yield.

### 3.2.6. Soil Drainability

Soil drainability is an essential factor in tea cultivation, as it affects the growth and yield of plants [36]. Drainage is necessary for tea cultivation because tea plants require well-drained soil to thrive. Tea plants are susceptible to root rot and other soil-borne diseases if they are grown in poorly drained soils, which can ultimately reduce the yield and quality of the tea produced. Tea plants grown in well-drained soils (Figure 3B) show significantly better growth and yield than those grown in poorly drained soils. A previous study also found that soil drainage has a significant impact on the nutrient uptake of tea plants. Plants grown in well-drained soils have higher nutrient uptake and show better

leaf quality than those grown in poorly drained soils [37]. Therefore, tea growers need to ensure that their soil has adequate drainage before planting tea.

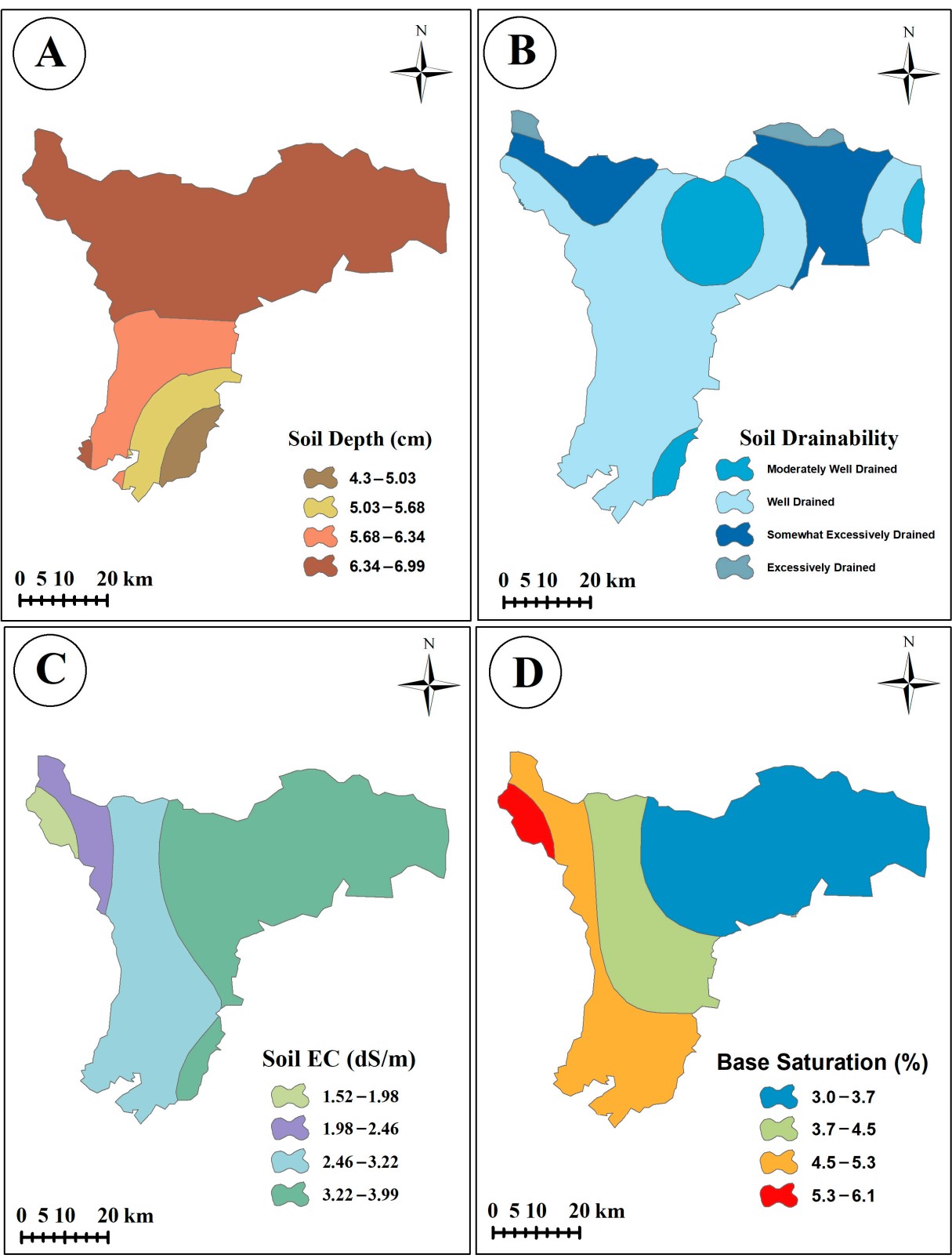

**Figure 3.** Suitability thematic layers for tea plantations: (**A**) soil depth, (**B**) soil drainability, (**C**) soil electrical conductivity, and (**D**) base saturation.

### 3.2.7. Soil Electrical Conductivity (EC)

Soil electrical conductivity (EC) is an important parameter that affects the growth and development of tea plants. It is a measure of the ability of the soil to conduct electricity, which is influenced by factors such as soil texture, moisture content, and the presence of minerals and salts. Tea plants are particularly sensitive to changes in soil EC, as high levels of salinity can lead to reduced growth, lower yields, and poor quality of the harvested leaves [38]. High soil EC levels (Figure 3C) harm tea plant growth, particularly in terms of reducing shoot length and leaf area. Tea plants grown in soil with high EC have lower photosynthetic rates and chlorophyll content, which can ultimately lead to lower tea quality [39]. Farmers and growers need to carefully monitor soil EC levels and take appropriate measures to prevent excessive salinization of the soil, such as reducing the use of fertilizers and irrigation water with high salt content.

### 3.2.8. Soil Base Saturation

Soil base saturation is a crucial factor in tea cultivation, as it determines the availability of essential nutrients and influences soil pH. The base saturation refers to the relative allocation of basic cations within the cation exchange capacity (CEC) of the soil, such as calcium, magnesium, and potassium. In tea cultivation, a base saturation level of around 50–70% is considered ideal for optimal growth and productivity [40]. Soil base saturation (Figure 3D) significantly influences tea yield and quality. A previous study was conducted in the Darjeeling hills of India, where tea cultivation is a major economic activity. The researchers found that tea plants grown in soils with base saturation levels between 50–70% had higher leaf yield, better quality, and higher levels of important biochemical compounds such as catechins and caffeine. On the other hand, tea plants grown in soils with low base saturation levels (<50%) showed reduced growth, low yield, and poor quality.

### 3.2.9. Soil Texture

Soil texture is one of the most important factors affecting tea cultivation. The texture of the soil can greatly influence the growth, yield, and quality of tea plants. Tea plants require well-drained, deep soils that are rich in organic matter and have good water-holding capacity. The texture of soil also affects the availability of nutrients and the activity of soil microorganisms that are essential for plant growth. Tea plants grown in sandy loam soils have higher shoot growth and yield than those grown in clay soils [41]. The sandy loam soils (Figure 4A) also have higher levels of organic matter and available nitrogen, phosphorus, and potassium. Tea plants grown in sandy soils have higher levels of catechins and caffeine than those grown in clay soils. Therefore, soil with a sandy loam texture is considered ideal for tea cultivation, as it provides good drainage, good water-holding capacity, and good nutrient availability.

### 3.2.10. Soil pH

Soil pH is a critical factor in tea cultivation, as it influences the availability of essential nutrients to the tea plant, affecting its growth, yield, and quality. The significant effect of soil pH on the growth, yield, and quality of tea plants grown in soil with a pH range of 4.5 to 5.5 produces higher yields and better-quality tea than those grown in alkaline soils (Figure 4B) with a pH range of 6.0 to 7.0 [42]. Furthermore, a previous study suggested that maintaining optimal soil pH is crucial for sustainable tea production, as it not only improves plant growth and yield but also reduces the use of chemical fertilizers and pesticides, leading to a more environmentally friendly approach to tea cultivation.

### 3.2.11. Normalized Difference Vegetation Index (NDVI)

The normalized difference vegetation index (NDVI) is widely used for monitoring and assessing vegetation health, productivity, and stress (Figure 4C). Tea cultivation is a major agricultural activity in many countries, and the NDVI is a useful tool for monitoring and managing tea plantations. Several studies have highlighted the importance of the NDVI

in tea cultivation [39]. For instance, a study conducted in China found that the NDVI can effectively monitor the spatial–temporal variability of tea growth and yield. Another study, conducted in India, showed that the NDVI can be used to assess the effect of different nutrient management practices on tea growth and yield.

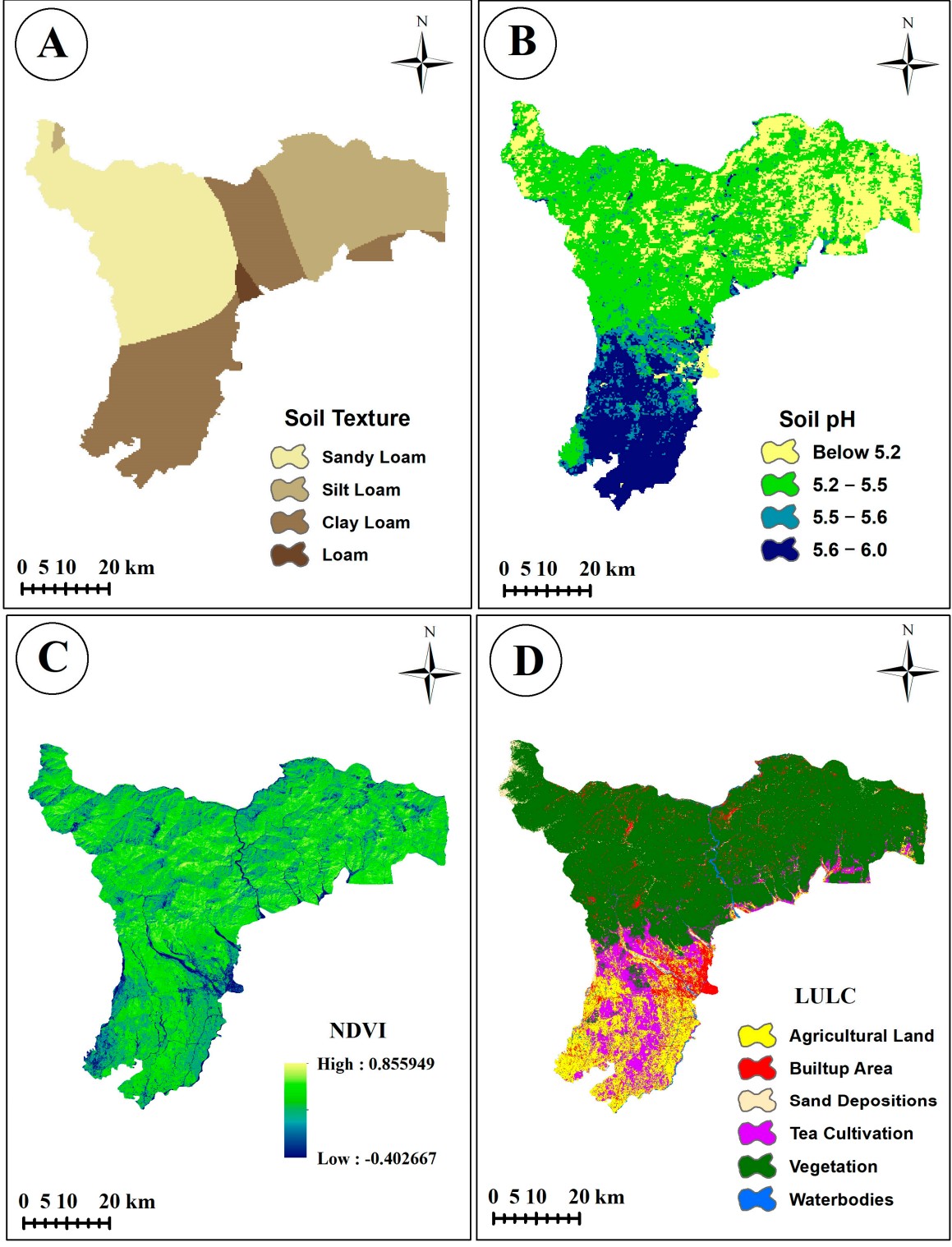

**Figure 4.** Suitability thematic layers for tea plantations: (**A**) soil texture, (**B**) soil pH, (**C**) NDVI, and (**D**) LULC.

### 3.2.12. Land Use/Land Cover (LULC)

Land use/land cover (LULC) mapping is an important tool for understanding the distribution and extent of different land cover types, including tea cultivation. Tea cultivation is a major land use in many countries and LULC mapping can provide insights into the spatial distribution of tea plantations, as well as their environmental and socioeconomic impacts. Several studies have highlighted the importance of LULC mapping in tea cultivation. LULC mapping (Figure 4D) can help to identify areas where tea plantations are expanding and encroaching on forested areas [43]. Another study conducted in India showed that LULC mapping can help to assess the impact of different land use types on tea productivity and quality [44].

LULC mapping is an important tool for tea farmers and researchers to understand the spatial distribution and impact of tea cultivation. Its applications in tea cultivation have been widely documented and it has been shown to have practical benefits in improving tea productivity and sustainability.

### 3.3. Methods

#### 3.3.1. Logistic Regression Model (LR)

The logistic regression (LR) model is a statistical approach used to model the relationship between an independent variable and a dependent variable [45,46]. This technique is commonly used in classification problems, where the goal is to predict the probability of a binary outcome based on one or more input variables. It is a supervised learning method that is widely applied in various fields. Logistic regression is categorized into two main types: simple logistic regression and multivariable logistic regression. Simple logistic regression involves only one predictor variable, while multivariable logistic regression is applied when there are several predictors, including both categorical and continuous variables. Logistic regression is a popular technique for binary classification problems and is widely used in the field of statistical modelling [47].

Using a technique called maximum likelihood estimation, the logistic regression model is trained by identifying the coefficient values that maximize the probability of examining the training data. Once trained, the model may be used for predicting the likelihood of an outcome for new input data by simply entering the values of the input variables into the model equation and using the logistic function. The benefit of logistic regression is that, unlike traditional linear regression, where the variables must all have normal distributions, the variables can be continuous, discrete, or any combination of the two, and they are not required to have normal distributions. This is accomplished by adding an appropriate link function to the standard linear regression model. In our study, the binary dependent variable represents the presence or absence of a tea plantation. To analyze the possibility of tea plantation, the input value of the dependent variable must be either 0 or 1.

The equation below shows the relationship between the occurrence and its dependency on multiple variables:

$$p = 1 / \left(1 + e^{-z}\right) \tag{1}$$

where $p$ is the probability of an event occurring and $z$ is the linear combination. In our study, the value $p$ is the estimated probability of a suitable tea plantation site. The value of $p$ ranges from 0 to 1 on an S-shaped curve. Logical regression entails fitting an equation of the following kind to the data:

$$z = b_0 + b_1 x_1 + b_2 x_2 + b_3 x_3 + b_n x_n \tag{2}$$

where $b_0$ is the intercept of the model, the $b_i$ (0, 1, 2, ... , n) are the slope coefficients of the LR model, and the $x_i$ (0, 1, 2, ... , n) are the independent variables.

This model assessed the spatial relationship between tea cultivation samples and conditioning factors. The spatial databases of each factor were converted raster to ASCII format in ArcGIS 10.2 software for use in IBM SPSS V28 software. Therefore, logistic regression mathematical equations were used to formulate each factor and obtain the coef-

ficient values (Table 1). Finally, the probability assessment of tea plantation occurrence was determined by analyzing spatial data via mathematical equations. (Equations (1) and (2)). In addition, logistic regression mathematical equations (Equations (1) and (3)) were used for all factors to generate an outcome of tea-plantation-suitability zones. The coefficient values of different conditioning factors were used for mapping tea-cultivation-suitability zones in ArcGIS 10.2, based on Equation (3) in the raster calculator (Figure 5).

$$
\begin{aligned}
z_p = {} & (0.001798 \times Rainfall) + (0.000295 \times Temperature) + (-0.07205 \times Elevation) + (0.04237 \times Slope) + Depth_c \\
& + Drainability_c + Ec_c + Saturation_c + Texture_c + pH_c + (0.024655 \times NDVI) + LULC_c - 17.640
\end{aligned} \tag{3}
$$

where *Depth* is soil depth, *Drainability* is soil drainability, $Ec$ is soil electrical conductivity, *Saturation* is base saturation, *Texture* is soil texture, *pH* is soil pH, and the coefficient values are listed in Table 2. A positive coefficient indicates that the tea plantation is more suitable at the level of a predictor than at the reference level of the factors. A negative coefficient indicates that the tea plantation is less suitable at the level of predictors than at the reference level of the factors (Table 2).

**Table 2.** Coefficient values of different discrete data factors.

| Factors | Class | No. of Pixels in a Class | % of Pixels in a Class | No. of Pixels of Tea Cultivation | % of Pixels of Tea Cultivation | Coefficient of Logistic Regression |
|---|---|---|---|---|---|---|
| Soil depth | 4.3–5.03 | 1091 | 3.66 | 268 | 9.69 | 0.29 |
| | 5.03–5.68 | 2519 | 8.46 | 729 | 26.37 | 0.477 |
| | 5.68–6.34 | 4738 | 15.91 | 1343 | 48.57 | 0.462 |
| | 6.34–6.99 | 21,434 | 71.97 | 425 | 15.37 | 0.306 |
| Soil texture | Sandy loam | 11,286 | 37.90 | 495 | 17.92 | 0.07 |
| | Silt loam | 7622 | 25.60 | 166 | 6.01 | 0.190 |
| | Clay loam | 10,581 | 35.54 | 2082 | 75.35 | 0.412 |
| | Loam | 287 | 0.96 | 20 | 0.72 | 0.057 |
| Soil pH | below 5.2 | 267 | 0.90 | 0 | 0.00 | −0.76 |
| | 5.2–5.5 | 15,122 | 50.82 | 163 | 5.89 | −1.01 |
| | 5.5–5.6 | 7085 | 23.81 | 585 | 21.15 | −0.34 |
| | 5.6–6.0 | 7281 | 24.47 | 2018 | 72.96 | 0.09 |
| Soil drainability | Moderately well-drained | 4570 | 15.36 | 169 | 6.11 | 0.745 |
| | Well-drained | 18,024 | 60.57 | 2465 | 89.12 | 1.096 |
| | Somewhat excessively drained | 6442 | 21.65 | 132 | 0.44 | 1.145 |
| | Excessively drained | 720 | 2.42 | 0 | 0.00 | 0.866 |
| Soil electrical conductivity | 1.52–1.98 | 786 | 2.64 | 0 | 0.00 | 0.172 |
| | 1.98–2.46 | 1963 | 6.60 | 543 | 19.63 | 0.069 |
| | 2.46–3.22 | 10,265 | 34.49 | 2105 | 76.10 | 0.016 |
| | 3.22–3.99 | 16,734 | 56.25 | 118 | 4.27 | 0.006 |
| Soil base saturation | 3.0–3.7 | 14,062 | 47.27 | 350 | 12.65 | −0.501 |
| | 3.7–4.5 | 6981 | 23.46 | 955 | 34.53 | 14.021 |
| | 4.5–5.3 | 7861 | 26.42 | 1461 | 52.82 | 14.593 |
| | 5.3–6.1 | 847 | 2.85 | 0 | 0.00 | 14.238 |
| LULC | Agricultural land | 560 | 1.88 | 0 | 0.00 | 0.722 |
| | Built-up area | 344 | 1.16 | 0 | 0.00 | −0.567 |
| | Sand depositions | 1636 | 5.50 | 0 | 0.00 | 0.019 |
| | Tea cultivation | 2990 | 10.05 | 2770 | 100.00 | 1.381 |
| | Vegetation | 20,288 | 68.21 | 0 | 0.00 | 0.206 |
| | Waterbodies | 3927 | 13.20 | 0 | 0.00 | −0.410 |

Coefficient of logistic regression value for continuous data: temperature 0.000295, slope 0.04237, rainfall 0.001798, NDVI 0.024655, and elevation −0.07205.

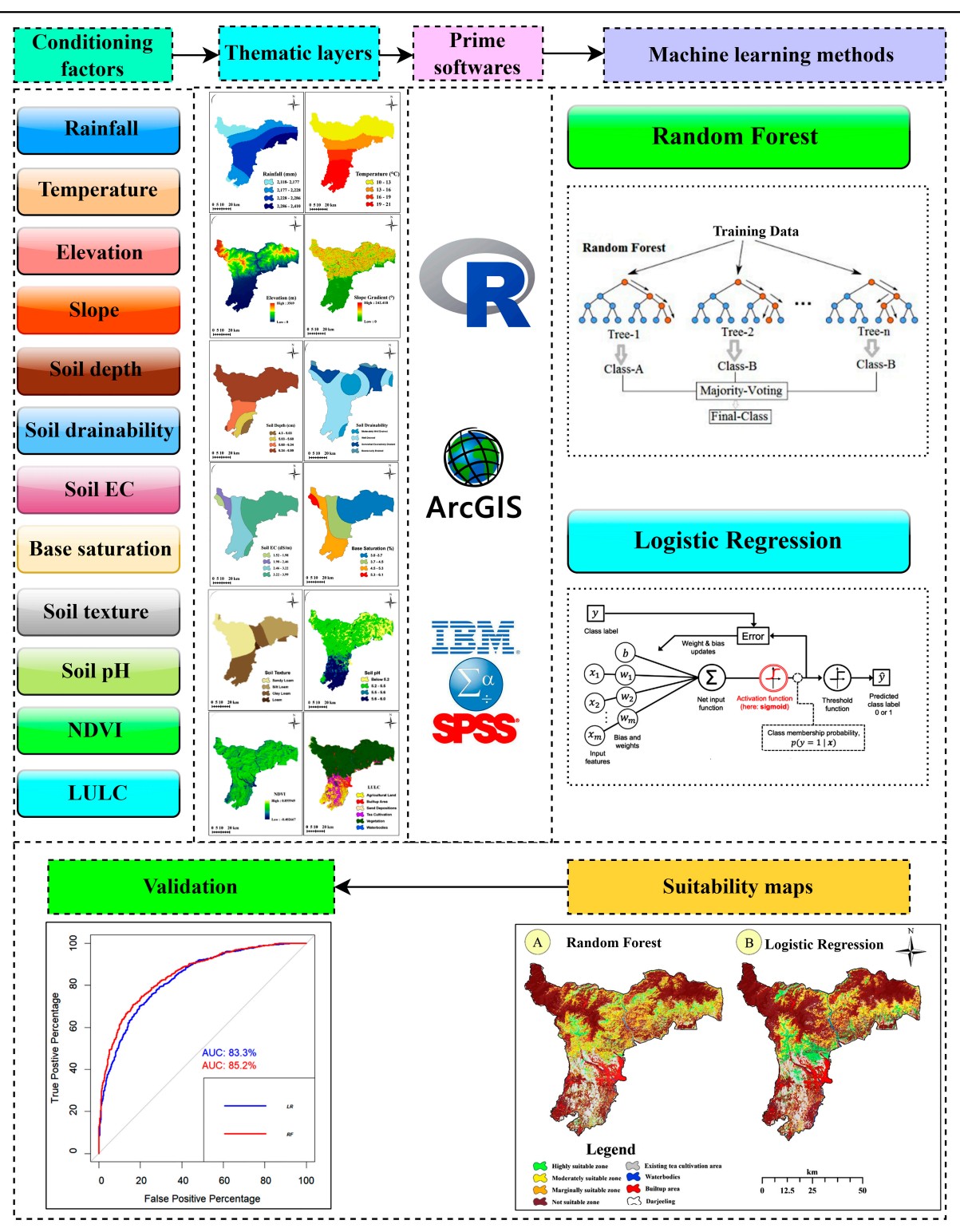

**Figure 5.** Methodological flowchart of this study.

### 3.3.2. Random Forest (RF)

The random forest (RF) model is widely recognized as a prominent algorithm in the domain of supervised machine learning, suitable for addressing classification as well as regression tasks [48,49]. The algorithm constructs decision trees by leveraging multiple instances, employing majority voting for classification and averaging values for regression

tasks. An important feature of RF is its ability to handle datasets with both continuous and categorical variables for both regression and classification tasks. RF has been shown to achieve excellent performance in classification problems. This model is also very useful for remote-sensing data classification [50]. The random forest approach [51] was developed by Leo Breiman; it is a compilation of unpruned decision trees designed for classification or regression purposes. These trees are built from randomly picked training data samples, and random features are chosen during the induction process. To make a prediction, the ensemble integrates the predictions of its members using a summing method. This typically involves taking the majority vote of the ensemble's members for classification tasks, or the average prediction for regression tasks [52].

In this study, the tea-cultivation status of the study area was collected via the tea-cultivation and non-tea-cultivation field investigation and the remote sensing techniques. The tea-cultivation land value was denoted as 1 and the non-tea-cultivation land as 0 in ArcMap 10.2. The values from all variables were collected using the "multi values to point" algorithm. R statistical software provides limited tuning options for random forests [53,54], including "ntree", "mytr", "sampsize", "nodsize", and "maxnodes". The "ntree'"parameter is used to specify the number of trees to be used in the algorithm. Using a proper number of trees can help to reduce errors, but using an excessive number of trees might be inefficient, specifically when dealing with large datasets. The "mytr" is the number of variables picked at random as candidates at each split. When performing classification, a vector called "sampsize" can be used to specify the number of samples to be drawn from each stratum during stratified sampling. The vector's length should correspond to the number of strata, and each element represents the desired sample size for the corresponding stratum. In this study, "ntree" is 100 number of trees and the "mytr" variable sample split is 1 [55]. The data were split randomly, where 70% of the data were used as training data to calibrate the model and the remaining 30% of the data were used as testing data for model validation. Finally, we stacked all the variables as a raster layer and prepared it for the final output (Figure 5).

### 3.4. ROC (Receiver Operating Characteristic) Curve

The receiver operating characteristic (ROC) curve is a visual representation of a classification model's performance across various classification thresholds. It plots the relationship between the true positive rate (*TPR*) (Equation (4)) and the false positive rate (*FPR*) (Equation (5)). *TPR* is computed as the ratio of true positives (*TP*) to the sum of true positives and false negatives (*FN*) using Equation (4), while *FPR* is calculated as the ratio of false positives (*FP*) to the sum of false positives and true negatives (*TN*) using Equation (5).

$$TPR = \frac{TP}{TP + FN} \tag{4}$$

$$FPR = \frac{FP}{FP + TN} \tag{5}$$

In these equations, *TP* represents acceptably classified positive cases, *TN* represents correctly classified negative cases, *FP* represents wrongly classified negative instances, and *FN* represents wrongly classified positive instances.

The area under the ROC curve (AUC) is a commonly used metric for evaluating the performance of a classification model. AUC values range between 0 and 1, where higher values show better performance. AUC values from 0.9 to 1 are considered outstanding, values from 0.8 to 0.9 are considered excellent, values from 0.7 to 0.8 are considered fair, values from 0.6 to 0.7 are considered poor, and values from 0.5 to 0.6 are considered failed. The AUC provides a comprehensive measure of the model's capability to discriminate between positive and negative instances across different classification thresholds [56,57].

## 4. Results

The study aimed to map the tea-plantation-suitability zones (TPSZs) in a particular region and to compare the performance of two different models, random forest and logistic regression, in predicting the suitability of an area for tea plantation. In this section, we elaborate on the results obtained in the study, providing a comprehensive analysis of the study findings. To begin with, the study collected data on various factors that are essential for tea plantations, including topography, climatic conditions, soil characteristics, and land use. There are six factors that play crucial roles, i.e., Lulc, NDVI, elevation, temperature, slope. and soil pH, for this tea-plantation-suitability-zone analysis (Figure S1). The data were collected from different sources, including satellite imagery, ground surveys, and meteorological data records. The collected data were then preprocessed to ensure that they were clean, accurate, and ready for analysis. To ensure that the random forest and logistic regression models were accurately calibrated, all factors were adjusted to the same scale and extension. The layers were prepared using ArcGIS 10.2, and the random forest model utilized R-4.3.0 statistical software, while the logistic regression model used IBM SPSS V28 software.

The study involved using the calibrated models to prepare a map of the tea-plantation-suitability zones (TPSZs). The TPSZs were divided into four categories—highly suitable zones, moderately suitable zones, marginally suitable zones, and not-suitable zones (Figure 6)—based on the predicted suitability values. The study involved using the calibrated models to map the TPSZs in the study area.

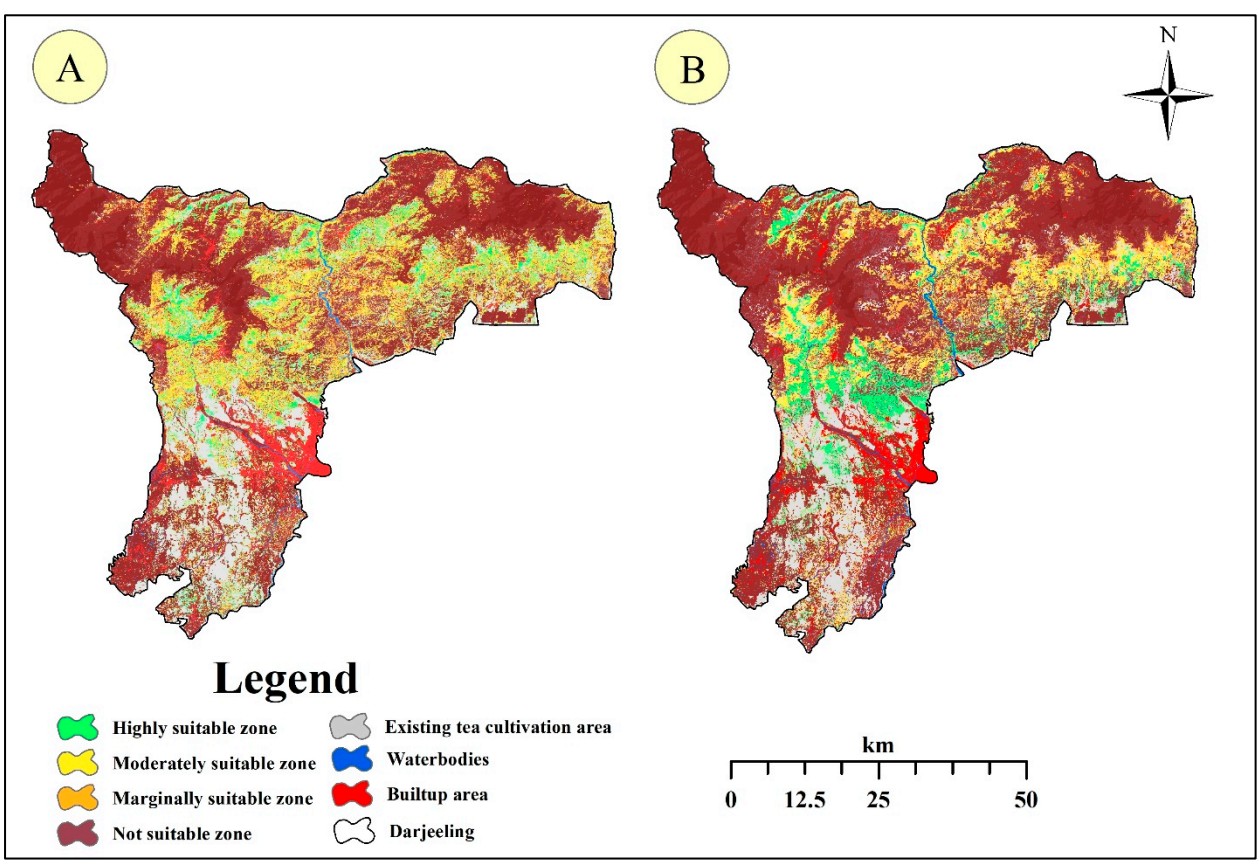

**Figure 6.** Tea-cultivation site suitability zones using random forest model (**A**) and logistic regression (**B**).

### 4.1. Tea-Plantation-Suitability Zones (TPSZs)

It was found that the total area covered by the TPSZs was 3149 km². The TPSZs were divided into four categories—highly suitable zones, moderately suitable zones, marginally suitable zones, and not-suitable zones (Figure 6)—based on the predicted suitability values.

The highly suitable zones covered 17.59% (553.84 km$^2$) for the random forest model and 20.87% (657.14 km$^2$) for the logistic regression model. The moderately suitable zones covered 16.59% (522.29 km$^2$) for the random forest model and 28.74% (905.11 km$^2$) for the logistic regression model. The marginally suitable zones covered 13.35% (420.47 km$^2$) for the random forest model and 15.80% (497.58 km$^2$) for the logistic regression model. The not-suitable zones covered 52.47% (1652.40 km$^2$) for the random forest model and 34.59% (1089.17 km$^2$) for the logistic regression model (Figure 7).

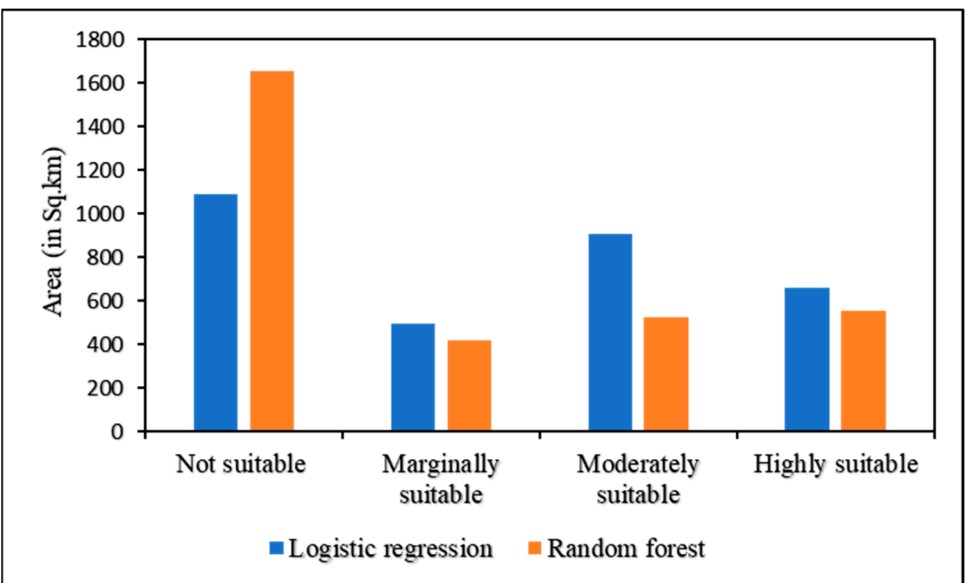

**Figure 7.** Area distribution of tea-plantation-suitability zones.

The study also revealed that the highly suitable and moderately suitable zones covered a significant proportion of the TPSZs, indicating that these areas have high potential for tea plantation. These zones could be prioritized for tea-plantation development, as they are likely to yield higher productivity and profitability. On the other hand, the marginally suitable and not-suitable zones may not be suitable for tea plantation development and, therefore, alternative land use options should be explored in these areas.

*4.2. Accuracy Assessment of Models*

The sample data were divided into two sets: 70% for model calibration and 30% for model validation. The models were calibrated using the training data and the accuracy of the models was assessed using the testing data. The performance of the models was measured using the area under the c(AUC) of the receiver operating characteristic (ROC) curve, which is a standard metric for assessing the performance of binary classifiers. The results showed that both models were accurate in predicting the suitability of an area for tea plantation, with AUC values of 85.2% and 83.3% for the random forest and logistic regression models, respectively. The Kappa value was 0.6916 and the confusion matrix (see the Supplementary Information for confusion statistics) showed that in the predicted map of tea plantation, positive occurrence points accurately showed 650 points for tea plantation and 865 points for no-tea plantation and negative occurrence points showed 197 points for tea plantation and 76 points for no-tea plantation (Figure 8). However, the random forest model performed better than the logistic regression model in terms of accuracy and precision, as indicated by the higher AUC value (Figure 8).

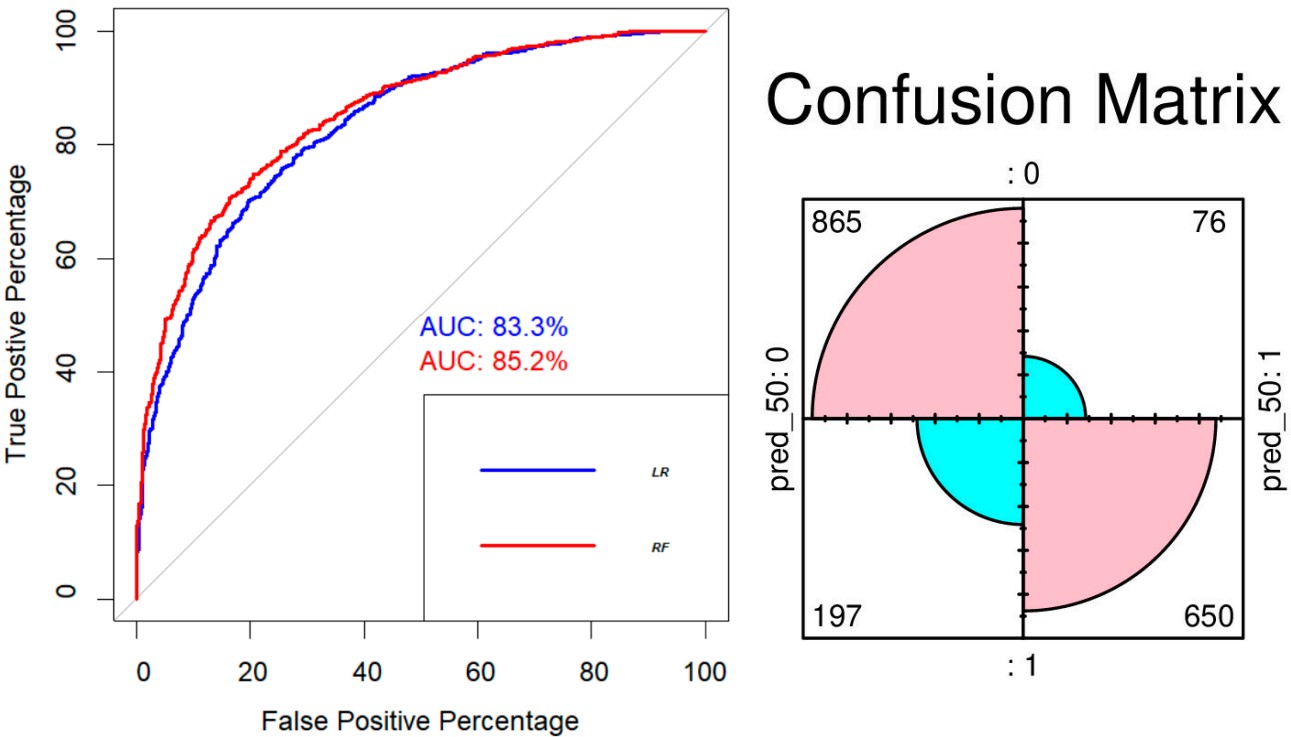

**Figure 8.** Receiver operating characteristic curve and confusion matrix.

### 4.3. Intercomparison of Models for Tea-Plantation Suitability

This study's findings provide valuable insights into the suitability of an area for tea plantation and the performance of two different models in predicting such suitability. The results showed that both models were accurate in predicting the suitability of an area for tea plantation, but the random forest model performed better than the logistic regression model. The ROC curve serves as validation for the suitability maps, and our findings indicate that the random forest model yielded better results than the logistic regression model, with an improvement in accuracy of 1.9%. The comparatively good performance of random forest could be attributed to the fact that random forest is an ensemble method that combines multiple decision trees, thereby improving the accuracy and robustness of the model. The study highlights the high suitability of non-forest areas for tea plantations but emphasizes the need to consider the potential positive and negative impacts on both the environment and the economy.

This study's results also highlight the importance of data preprocessing and calibration in ensuring the accuracy and reliability of the models. The collected data were preprocessed to ensure that they were clean, accurate, and ready for analysis. The models were then calibrated using the calibration dataset, which helped to optimize the models' parameters and improve their performances. The validation dataset was used to assess the accuracy of the models, and the AUC values showed that the models were accurate in predicting the suitability of an area for tea plantation.

## 5. Discussion

In this study, we utilized both random forest and logistic regression models to identify suitable sites for tea plantations. We conducted extensive mapping of 2770 tea plantation locations using a combination of satellite imagery and field surveys. Both the random forest and logistic regression models were employed to analyze the TPSZs, using the aforementioned 12 factors. Before the analysis, we standardized the cell size and row column to ensure an equal number of data points. We further normalized the data using ArcGIS 10.2. The discrete data layer was subsequently classified based on the tea plantation site, and both models adopted a grid-based analysis methodology.

The climate plays a pivotal role in influencing the distribution, growth, and yield of tea plantations. Optimal tea cultivation occurs in warm and moist climatic conditions, characterized by a consistent temperature throughout the year. Adequate and well-distributed annual rainfall, typically ranging from 1500 to 2000 mm, is necessary for successful tea cultivation. However, extended dry seasons, excessively low temperatures (below 17.5 °C), or extremely high temperatures (above 26.5 °C) can have detrimental effects on tea plant foliage in the Darjeeling Himalaya [58]. Consequently, tea cultivation has been facing serious issues posed by inadequate water supply and large dam construction [2]. Furthermore, the increasing intensity of temperature also impacts crucial biological processes such as photosynthesis and reproductive growth in tea plantations [30,31].

Soil characteristics play a critical role in tea plantations. The physical, chemical, and biological properties of the soil collectively contribute to the success of tea plantations. Notably, soil texture and pH emerged as the most significant physicochemical factors, with an acidic soil pH ranging between 5.6 and 6.0 being deemed ideal for tea cultivation in the study area. However, in recent decades, increasing soil pH along with sediment-based soil texture has created a barrier for the quality of tea plantation in the low-elevated areas of northwestern Darjeeling Himalaya. Moreover, base saturation (i.e., Ca, Mg, K, and Na) has been found to be less in the northeast section and, consequently, the quality of tea production was also somewhat damaged. Similar kinds of results have been found in Bangladesh tea gardens [40] and in Malaysia [59,60]. Regarding drainability, soils with a sandy or coarser grain texture exhibit superior permeability and optimal drainability; they are found in the low-elevated areas. except in the northwestern and northeastern sections. Therefore, comparatively low-elevated but gentle-to-moderate slope areas provide optimal conditions for tea growth, and the northwestern and northeastern sections are comparatively less productive for tea growth.

However, the current land-use scenario—specifically, agriculture and vegetation status—were determined where are favorable conditions for tea growth [2]. The analysis revealed that the dominant LULC—specifically, agricultural land and vegetation cover—have sometimes created a problem in tea plantations. Dense forest cover or intensive agriculture activities near tea gardens consume more water, and recharge in the ground creates a problem for the growth of tea.

This study has some limitations. The RF model tends to overfit the training data if not properly tuned. Overfitting can lead to misleading predictions and reduced model performance. Both the RF and LR models require a sufficient amount of high-quality data for accurate prediction. The RF and LR models are best suited for site suitability analysis and can be used in different studies, such as studies of flood susceptibility, landslide susceptibility, forest fire susceptibility, landfill suitability, and urban built-up suitability [61].

This study was able to identify substantial potential areas for expanding tea plantations in Darjeeling Himalaya, based on suitable surface and environmental conditions such as slope, rainfall, and temperature. Moreover, the soil conditions in the study area were found to be conducive to tea plantations.

## 6. Conclusions

This study employed random forest (RF) and logistic regression (LR) models to assess the suitability of tea-plantation sites based on several factors, including temperature, rainfall, elevation, soil characteristics, and land use/land cover. The models were calibrated and validated using standardized data, and GIS techniques were employed to map tea-plantation-suitability zones (TPSZs).

The study found that both the RF and LR models were effective in classifying TPSZs into four categories: highly suitable zones, moderately suitable zones, marginally suitable zones, and not-suitable zones. The total area covered by the TPSZs was 3149 km$^2$. The RF model identified 17.59% of the total area as highly suitable, 16.59% as moderately suitable, 13.35% as marginally suitable, and 52.47% as not suitable. The LR model identified 20.87% of the total area as highly suitable, 28.74% as moderately suitable, 15.80% as marginally

suitable, and 34.59% as not suitable. The study used 2770 sample points for TPSZ mapping, with 70% of the locations used for model calibration and 30% for validation. The area under the curve (AUC) for the RF model was 85.2%, while for the LR model, it was 83.3%. These high AUC values indicated the accuracy and justification of the models in predicting tea-plantation potential.

The study found that climate factors such as temperature and rainfall played a crucial role in determining tea-plantation suitability. Soil characteristics, including soil depth, texture, pH, drainability, electrical conductivity, and base saturation, were also significant determinants. The analysis of land use/land cover using the NDVI and LULC indices highlighted the dominant presence of forest cover and tea plantations in the study area. The RF model outperformed the LR model, demonstrating a 1.9% improvement in accuracy. The suitability maps generated by the RF model provided valuable insights into tea-plantation potential, while considering the balance between environmental and economic impacts. This study recommends the protection of natural reserve areas, while promoting tea plantation expansion in non-forest regions.

In summary, the RF and LR models facilitated the identification and classification of suitable tea-plantation sites based on multiple factors. The study's findings can assist stakeholders in making informed decisions regarding tea-plantation site selection and management, while considering environmental and economic aspects.

**Supplementary Materials:** The following supporting information can be downloaded at: https://www.mdpi.com/article/10.3390/su151310101/s1, Figure S1: Importance of different factors for Tea plantation suitability analysis.

**Author Contributions:** Conceptualization, N.S.; data curation, P.D., A.S. and A.V.; formal analysis, N.S., P.D. and A.V.; funding acquisition., N.S., A.S. and S.K.M.; methodology, N.S. and P.D.; supervision, N.S. and R.N.; visualization, N.S., P.D., A.V., R.K. and A.K.; validation, P.D.; writing—original draft, P.D., A.V. and A.S.; project administration, N.S. and R.N.; writing—review and editing, N.S., A.S.; S.K.M., R.N., S.P.A. and B.P. All authors have read and agreed to the published version of the manuscript.

**Funding:** This research was supported by the Teaching Learning Centre, Ramanujan College, New Delhi, India, (project code: MP02/2022-23). The mentioned project is funded under the PMMMNMTT scheme, Department of Higher Education, Ministry of Education, Government of India.

**Institutional Review Board Statement:** Not applicable.

**Informed Consent Statement:** Not applicable.

**Data Availability Statement:** Publicly available datasets were analyzed in this study. Analyzed data are available on request from the corresponding author.

**Acknowledgments:** The authors acknowledge the data shared by the Climate Research Unit, University of East Anglia; Google Earth Engine; R Project; BHUVAN, the Indian Space Research Organisation; the International Soil Reference and Information Centre; and the National Aeronautics and Space Administration, USA. We are thankful to the Teaching Learning Centre, Ramanujan College, New Delhi, India, for the major project (project code: MP02/2022-23). The mentioned project is funded under the PMMMNMTT scheme, Department of Higher Education, Ministry of Education, Government of India.

**Conflicts of Interest:** The authors declare no conflict of interest.

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
