# Peer review of "Analysis of Tea Plantation Suitability Using Geostatistical and Machine Learning Techniques: A Case of Darjeeling Himalaya, India"

_sustainability, doi:10.3390/su151310101_

Round 1

Reviewer 1 Report

There are a couple of lapses from correct English grammar.

See above.

Author Response

Reviewer #1:

  1. There are a couple of lapses from correct English grammar.

Author’s response: Thank you so much for your valuable comment. The authors have revised the entire manuscript and corrected the English grammar by native anonymous.

Reviewer 2 Report

This paper identified suitable sites for tea plantations considering twelve important conditioning factors using logistic and random forest regression models, which is a very practical and excellent view of research. The findings of this paper can help the stakeholders in making decisions regarding tea plantation. However, in my view, although the paper indicated that both climate factors and soil characteristics played a crucial role in determining tea plantation, it does not explore what the dominant driving factors are under the classification results. In addition, the review of previous methods can be condensed further and the innovation point of the article needs to be further highlighted. In the discussion part, the second and third paragraphs seem to repeat the importance of meteorological soil and other factors in the Method 3.2. Please revise them further. Overall, this manuscript is well-written in language and structure, but a moderate revision is required to clarify innovative points of the article and modify the discussion part.

Question:

Could you add more historic years(not just 2022) to support your result? And could you add some discussion of explaining which soil and climate factors take leading role for the classification results?

There are some questions about the draft as listed:

Line86: there are two “were tested”, please modify that.

Line505: “4. Discussion” should be “5. Discussion”.

Line548: “5. conclusions” should be “6. conclusions”

This paper identified suitable sites for tea plantations considering twelve important conditioning factors using logistic and random forest regression models, which is a very practical and excellent view of research. The findings of this paper can help the stakeholders in making decisions regarding tea plantation. However, in my view, although the paper indicated that both climate factors and soil characteristics played a crucial role in determining tea plantation, it does not explore what the dominant driving factors are under the classification results. In addition, the review of previous methods can be condensed further and the innovation point of the article needs to be further highlighted. In the discussion part, the second and third paragraphs seem to repeat the importance of meteorological soil and other factors in the Method 3.2. Please revise them further. Overall, this manuscript is well-written in language and structure, but a moderate revision is required to clarify innovative points of the article and modify the discussion part.

Question:

Could you add more historic years(not just 2022) to support your result? And could you add some discussion of explaining which soil and climate factors take leading role for the classification results?

There are some questions about the draft as listed:

Line86: there are two “were tested”, please modify that.

Line505: “4. Discussion” should be “5. Discussion”.

Line548: “5. conclusions” should be “6. conclusions”

Author Response

Reviewer #2:

  1. In my view, although the paper indicated that both climate factors and soil characteristics played a crucial role in determining tea plantation, it does not explore what the dominant driving factors are under the classification results.

 Author’s response: Thank you so much for your valuable comment. The authors have now incorporated it in the result section of the revised manuscript. Please see the results section.

The following supplementary figure has also been added as Figure- S1.

  1. The review of previous methods can be condensed further and the innovation point of the article needs to be further highlighted.

Author’s response: Thank you so much for your valuable comment. The authors have revised the introduction section and incorporated the suggestion. Please see the introduction section.

  1. In the discussion part, the second and third paragraphs seem to repeat the importance of meteorological soil and other factors in the Method 3.2. Please revise them further.

Author’s response: Thank you so much for your valuable comment. The authors have revised the discussion section as per the suggestions. Please see the discussion section.

  1. Could you add more historic years (not just 2022) to support your result? And could you add some discussion of explaining which soil and climate factors take leading role for the classification results?

Author’s response: Thank you so much for your valuable comment. In this research article we attempted to find out the future tea plantation suitability zones in Darjeeling region.  We used recent data to calibrate the models and therefore there is no need to use historical data to support future result. The authors have also incorporated the discussion on leading role about factors and add figure in in supplementary file as figure S1.

  1. There are some questions about the draft as listed:

Line86: there are two “were tested”, please modify that.

Line505: “4. Discussion” should be “5. Discussion”.

Line548: “5. conclusions” should be “6. conclusions”

Author’s response: Thank you so much for your valuable observations. We made corrections at Line86, Line 505 and 548 respectively.

Reviewer 3 Report

This is an important research especially considering the scarcity of arable lands for cultivation and considering the perennial nature of tea plantations. The topic may not be ground braking pioneering research using this technology, however, it addresses a timely an important need.

1.      Darjeeling is an area in India – country name also should be indicated in the title and at the beginning of the introduction for the benefit of international readership.

2.      Need to expand unknown terms in the abstract and throughout at their first use (e.g. NDVI, LULC etc)

3.      The methodology section should be limited to explain the exact methodology that is followed in conducting the research. 

However, the authors have included a lot of extra sections that should have been included either in the introduction or discussion (one example is the environmental conditions suitable for tea cultivation). The methodology should be strictly limited to explain the research methodology used.

Author Response

Reviewer #3:

  1. Darjeeling is an area in India – country name also should be indicated in the title and at the beginning of the introduction for the benefit of international readership.

Author’s response: Thank you so much for your valuable comment. The authors have revised the title. Please see the Title.

  1. Need to expand unknown terms in the abstract and throughout at their first use (e.g. NDVI, LULC etc)

Author’s response: Thank you so much for your valuable comment. The authors have correct it. Please see the Abstract.

  1. The methodology section should be limited to explain the exact methodology that is followed in conducting the research. However, the authors have included a lot of extra sections that should have been included either in the introduction or discussion (one example is the environmental conditions suitable for tea cultivation). The methodology should be strictly limited to explain the research methodology used.

Author’s response: Thank you so much for your valuable suggestions. The authors have separated the methods section and addressed your queries properly in the modified manuscript. Please take a look into the entire manuscript.

Reviewer 4 Report

The study presents intriguing and valuable insights, making it highly relevant and beneficial to the readers of the MDPI publication. The research findings of this study indicated that the random forest model exhibited greater accuracy in comparison to the logistic regression model.

For introduction, the reviewer is suggesting enriching this section by including previous works where random forest technique has been used in different fields. For example: https://doi.org/10.1038/s41598-021-96872-w and https://doi.org/10.1080/19942060.2022.2126528 

4.2 Accuracy Assessment of Models: The authors are encouraged to improve this section.

Conclusion section: Very important section in the paper, please summarize the findings, highlight the limitations of the current study and give recommendations of future works. 

  •  

Author Response

Reviewer #4:

  1. 2 Accuracy Assessment of Models: The authors are encouraged to improve this section.

 Author’s response: Thank you so much for your valuable comment. The authors have revised this section by adding kappa value and predicted maps confusion matrix. Please see the Accuracy Assessment of Models.

  1. Conclusion section: Very important section in the paper, please summarize the findings, highlight the limitations of the current study and give recommendations of future works.

 Author’s response: Thank you so much for your valuable suggestions. The authors have revised this section accordingly. Please see the Conclusion Section.

Round 2

Reviewer 2 Report

This paper identified suitable sites for tea plantations considering twelve important conditioning factors using logistic and random forest regression models, which is a very practical and excellent view of research. The findings of this paper can help the stakeholders in making decisions regarding tea plantation. However, in my view, although the paper indicated that both climate factors and soil characteristics played a crucial role in determining tea plantation, it does not explore what the dominant driving factors are under the classification results. In addition, the review of previous methods can be condensed further and the innovation point of the article needs to be further highlighted. In the discussion part, the second and third paragraphs seem to repeat the importance of meteorological soil and other factors in the Method 3.2. Please revise them further. Overall, this manuscript is well-written in language and structure, but a moderate revision is required to clarify innovative points of the article and modify the discussion part.

Question:

Could you add more historic years(not just 2022) to support your result? And could you add some discussion of explaining which soil and climate factors take leading role for the classification results?

There are some questions about the draft as listed:

Line86: there are two “were tested”, please modify that.

Line505: “4. Discussion” should be “5. Discussion”.

Line548: “5. conclusions” should be “6. conclusions”

This paper identified suitable sites for tea plantations considering twelve important conditioning factors using logistic and random forest regression models, which is a very practical and excellent view of research. The findings of this paper can help the stakeholders in making decisions regarding tea plantation. However, in my view, although the paper indicated that both climate factors and soil characteristics played a crucial role in determining tea plantation, it does not explore what the dominant driving factors are under the classification results. In addition, the review of previous methods can be condensed further and the innovation point of the article needs to be further highlighted. In the discussion part, the second and third paragraphs seem to repeat the importance of meteorological soil and other factors in the Method 3.2. Please revise them further. Overall, this manuscript is well-written in language and structure, but a moderate revision is required to clarify innovative points of the article and modify the discussion part.

Question:

Could you add more historic years(not just 2022) to support your result? And could you add some discussion of explaining which soil and climate factors take leading role for the classification results?

There are some questions about the draft as listed:

Line86: there are two “were tested”, please modify that.

Line505: “4. Discussion” should be “5. Discussion”.

Line548: “5. conclusions” should be “6. conclusions”

Author Response

Reviewer #2:

This paper identified suitable sites for tea plantations considering twelve important conditioning factors using logistic and random forest regression models, which is a very practical and excellent view of research. The findings of this paper can help the stakeholders in making decisions regarding tea plantation. However, in my view, although the paper indicated that both climate factors and soil characteristics played a crucial role in determining tea plantation, it does not explore what the dominant driving factors are under the classification results. In addition, the review of previous methods can be condensed further and the innovation point of the article needs to be further highlighted. In the discussion part, the second and third paragraphs seem to repeat the importance of meteorological soil and other factors in the Method 3.2. Please revise them further. Overall, this manuscript is well-written in language and structure, but a moderate revision is required to clarify innovative points of the article and modify the discussion part.

  1. In my view, although the paper indicated that both climate factors and soil characteristics played a crucial role in determining tea plantation, it does not explore what the dominant driving factors are under the classification results.

 Author’s response: Thank you so much for your valuable comment. The authors have incorporated it in the result section. Please See in the result section.

We also add this figure in the supplementary as a Figure- S1.

  1. The review of previous methods can be condensed further and the innovation point of the article needs to be further highlighted.

Author’s response: Thank you so much for your valuable comment. The authors have revised the introduction and incorporate it. Please See in the introduction section.

  1. In the discussion part, the second and third paragraphs seem to repeat the importance of meteorological soil and other factors in the Method 3.2. Please revise them further.

Author’s response: Thank you so much for your valuable comment. The authors want to discuss the importance to know the individual importance of each factor for tea cultivation in the method section and in the discussion section authors discuss the importance of all factors in this study area as a summary. Please See in the Materials and Methods section and also in discussion section.

  1. Could you add more historic years (not just 2022) to support your result? And could you add some discussion of explaining which soil and climate factors take leading role for the classification results?

Author’s response: Thank you so much for your valuable comment. In this research article we try to find out the future tea plantation suitability zones in Darjeeling region.  We try to use recent data to calibrate the models. So, there is no need to use historical data to support future result. The authors have also incorporated the discussion on leading role about factors and add figure in in supplementary file as figure S1.

  1. There are some questions about the draft as listed:

Line86: there are two “were tested”, please modify that.

Line505: “4. Discussion” should be “5. Discussion”.

Line548: “5. conclusions” should be “6. conclusions”

Author’s response: Thank you so much for your valuable observations. We correct and highlight it at Line86 repetition of “were tested” please find it in the line number 101. We correct it at Line 551 “4. Discussion” be “5. Discussion” and line 609 “5. conclusions” be “6. conclusions”.

Reviewer 3 Report

1. Section 2: study area can be moved to the materials and methodology section.

2. Limitations in the study model (RF model) need not be included under conclusions. This section with recommendations can be moved to the discussion section.

Author Response

Reviewer #3:

  1. Section 2: study area can be moved to the materials and methodology section.

Author’s response: Thank you so much for your valuable comment. The authors have sincerely discussed about it and decided that the study area not include in the materials and methos section. So, it will remain as a different section.

  1. Limitations in the study model (RF model) need not be included under conclusions. This section with recommendations can be moved to the discussion section.

Author’s response: Thank you so much for your valuable comment. The authors have removed it from the conclusion section and add in the discussion section. Please See in the Discussion section.

Round 3

Reviewer 2 Report

No further comments

No further comments